# Mortality and morbidity of asthma and chronic obstructive pulmonary disease associated with ambient environment in metropolitans in Taiwan

Yasmin Zafirah[1☉], Yu-Kai Lin[2☉], Gerry Andhikaputra[1], Li-Wen Deng[1], Fung-Chang Sung[3,4,5‡], Yu-Chun Wang[1,6‡]*

1 Department of Environmental Engineering, College of Engineering, Chung Yuan Christian University, Zhongli, Taiwan, 2 Department of Health and Welfare, University of Taipei College of City Management, Taipei, Taiwan, 3 Management Office for Health Data, China Medical University Hospital, Taichung, Taiwan, 4 Department of Health Services Administration, China Medical University, Taichung, Taiwan, 5 Department of Food Nutrition and Health Biotechnology, Asia University, Taichung, Taiwan, 6 Research Center for Environmental Changes, Academia Sinica, Nankang, Taipei, Taiwan

☉ These authors contributed equally to this work.
‡ FCS and YCW also contributed equally to this work.
* ycwang@cycu.edu.tw, swingapple@gmail.com

## Abstract

### Background

This study investigated risks of mortality from and morbidity (emergency room visits (ERVs) and outpatient visits) of asthma and chronic obstructive pulmonary disease (COPD) associated with extreme temperatures, fine particulate matter ($PM_{2.5}$), and ozone ($O_3$) by sex, and age, from 2005 to 2016 in 6 metropolitan cities in Taiwan.

### Methods

The distributed lag non-linear model was employed to assess age (0–18, 19–39, 40–64, and 65 years and above), sex-cause-specific deaths, ERVs, and outpatient visits associated with extreme high (99th percentile) and low (5th percentile) temperatures and $PM_{2.5}$ and $O_3$ concentrations at 90th percentile. Random-effects meta-analysis was adopted to investigate cause-specific pooled relative risk (RR) and 95% confidence intervals (CI) for the whole studied areas.

### Results

Only the mortality risk of COPD in the elderly men was significantly associated with the extreme low temperatures. Exposure to the 90th percentile $PM_{2.5}$ was associated with outpatient visits for asthma in 0–18 years old boys [RR = 1.15 (95% CI: 1.09–1.22)]. Meanwhile, significant elevation of ERVs of asthma for females aged 40–64 years was associated with exposure to ozone, with the highest RR of 1.21 (95% CI: 1.05–1.39).

**Data Availability Statement:** All relevant data are within the paper and its Supporting Information files.

**Funding:** This study was also supported by grants from the Ministry of Science and Technology (MOST 109-2625-M-033-002-, MOST 108-2625-M-033-002- and MOST 106-2221-E-033-006-MY2) - Yu-Chun Wang The National Health Research Institutes (NHRI-107A1-EMCO-3617191 and NHRI-106A1-PDCO-3617191) - Yu-Chun Wang Taiwan Ministry of Health and Welfare Clinical Trial Center (MOHW109-TDU-B-212-114004)- Fung-Chang Sung Taiwan Environmental Protection Administration - Yu-Chun Wang MOST Clinical Trial Consortium for Stroke (MOST 109-2321-B-039-002) - Fung-Chang Sung Tseng-Lien Lin Foundation, Taichung, Taiwan - Fung-Chang Sung The National Health Research Institutes (MOHW105-TDU-M-212-113003) - Fung-Chang Sung.

**Competing interests:** The authors have declared that no competing interests exist.

## Conclusions

This study identified vulnerable subpopulations who were at risk to extreme events associated with ambient environments deserving further evaluation for adaptation.

## Introduction

Global warming is an abrupt climate change, commonly defined by the increase in frequency and intensity of extreme temperatures [1–3]. The extreme temperature events have gained public health concerns due to increased premature deaths and hospitalizations [4–6]. According to the Taiwan Climate Change Projection and Information Platform Project, the average temperature in Taiwan has increased by 1.1–1.6°C, greater than the average global increase of 0.8°C reported by the United Nations-affiliated Intergovernmental Panel on Climate Change [7].

The economic development, consumption of energy, and rising number of transportation usage are causing the broad range of air pollution levels which put a large number of people at risk [8–10]. The exposure to air pollution can trigger respiratory and cardiovascular disorders [11]. Several epidemiological studies have shown that fine particles ($PM_{2.5}$) and ozone ($O_3$) play a role in adverse health effects [12–14]. Nevertheless, the vulnerability to $PM_{2.5}$ and $O_3$ between ages groups have been scarcely reported [15]. A US study reported that ambient temperature is likely higher in urban areas than in rural areas. This condition is known as the urban heat island effect, which explains a positive relation between ambient temperature and urban land use intensity [16]. The presence of high traffic density can contribute to pollution levels in urban areas [17].

Respiratory diseases, including asthma and COPD, are known among the top causes of deaths worldwide [18]. According to the WHO, asthma is estimated to cause about 250,000 annual deaths worldwide and COPD will move from fifth leading cause of death in 2002, to fourth place in the ranking projected to 2030 worldwide [19]. Among respiratory diseases, the prevalence of asthma and chronic obstructive pulmonary disease (COPD) have increased to 15–20% in Taiwan [20, 21]. Several risk factors are believed to play the role of increased vulnerability of these respiratory disorders, including age, sex, physical activity, lifestyle [13, 22]. However, environmental factors such as air pollution and ambient temperature also have been associated with the chronic respiratory diseases [5, 23].

Health risks associated with temperature vary among cities and population susceptibility against extreme temperatures and adaptation capabilities [24]. Subpopulation, including young children, the elderly, and industrial workers may be at a higher risk due to increased biological sensitivity and different exposure patterns [10]. Children are more sensitive at a young age, because they have premature lung growth. Male population poses a slightly higher risk than female population. There are stronger effects for boys in early life and for girls in later childhood, which may vary by stage of life, exposure to ambient environment, and hormonal status [13].

It is fundamental to improve an adaptive strategy and public-health policies to account the full range of effects of extreme temperature and air pollution associated with the health risks. Moreover, a better understanding of the vulnerability varied by age, sex, and disease in association with extreme temperatures is required to estimate appropriate strategies for health consequences under climate-change scenarios. Taiwan is a subtropical island with hot and humid climate condition. More than 65% population residing in metropolitans areas in 2019 [25],

and 94.7% of families were equipped with air conditioners installation [26]. Hence, this study aims to investigate the temperature- and $PM_{2.5}$ and $O_3$- related health risks of asthma and COPD in metropolitan cities (i.e., Taipei City, New Taipei City, Taoyuan City, Taichung City, Tainan City, and Kaohsiung City) in Taiwan. We evaluated risks of mortality from and morbidity of these two disorders by age and sex.

## Materials and methods

This study focused on six major metropolitan cities in Taiwan, including New Taipei City, Taipei City, Taoyuan City, Taichung City, Tainan City, and Kaohsiung City, as study areas (Fig 1).

### Data source and statistical analysis

We used daily data of deaths from asthma and COPD and records of emergency room visits (ERVs) and outpatient visits in the National Health Insurance database of 2005–2916, obtained from the Ministry of Health and Welfare for this study. Since the end of 2004, 99% of population living in Taiwan have been covered in the insurance [27]. All methods were carried out in accordance with relevant guidelines and all protocols were approved by Taiwan National Health Research Institutes (code: EC1081103-F-E). All identification numbers of insured population were randomized into surrogate numbers for users to protect privacy. We used *9t*h and *10th Revision* of the *International Classification of Diseases* (ICD-9) and (ICD-10) to identify asthma [include acute bronchitis and chronic bronchitis (ICD-9 466,491, 493 and ICD-10 J45-J46)] and COPD (ICD-9 496 and ICD-10 J44). Refer to previous Asian's reports [28, 29], this study only selected ICD-10 J44 as COPD definition and assess the associations with ambient environment. We assembled daily records of deaths, ERVs, and outpatient visits by ambient environment conditions for data analysis. We stratified ages into two groups: 40–64 and 65 years and above. The association between mortality and daily mean temperatures was only displayed for subpopulation aged 65 years and above because over 80% deaths occurred in this subpopulation. In addition, due to younger subpopulation is prone to comorbid with asthma disease in Taiwan [13], this study also evaluated associations between temperature and ERVs and outpatient visits of asthma of subpopulations aged 0–18 and 19–39 years.

From the Taiwan Central Weather Bureau, we retrieved daily weather data of temperature (°C), relative humidity (%), and wind speed (m/s), measured at 6 selected stations (e.g., Banqiao of New Taipei (station number: 466880), Taipei (station number: 466920), Taoyuan (station number: 46686), Taichung (station number: 467490 and 467770), Tainan, (station number: 467410 and 467420), and Kaohsiung (station number: 467440)).

Taiwan Air Quality Monitoring Network provided daily records of air pollutants. We used 24-h average concentration of fine particulates matter with diameters of less than 2.5 micron ($PM_{2.5}$) and ozone ($O_3$) in this study. Refer to our previous study [30], 24-h average concentration of ozone is the preferred metric instead of 1-h maximum concentration and 8-h maximum concentration of ozone for assessing the association with outpatient visits for total respiratory diseases. Detailed information on the monitoring instruments, stations, and quality assurance criteria are available on the webpage (https://airtw.epa.gov.tw/).

### Non-linear association between daily ambient environment and health risks

This study applied the distributed lag non-linear model (DLNM) with a quasi-Poisson regression, proposed by Gasparrini et al. (2017), to evaluate non-linear and lagged dependencies of exposure-response relationship between the ambient temperatures and deaths, ERVs, and

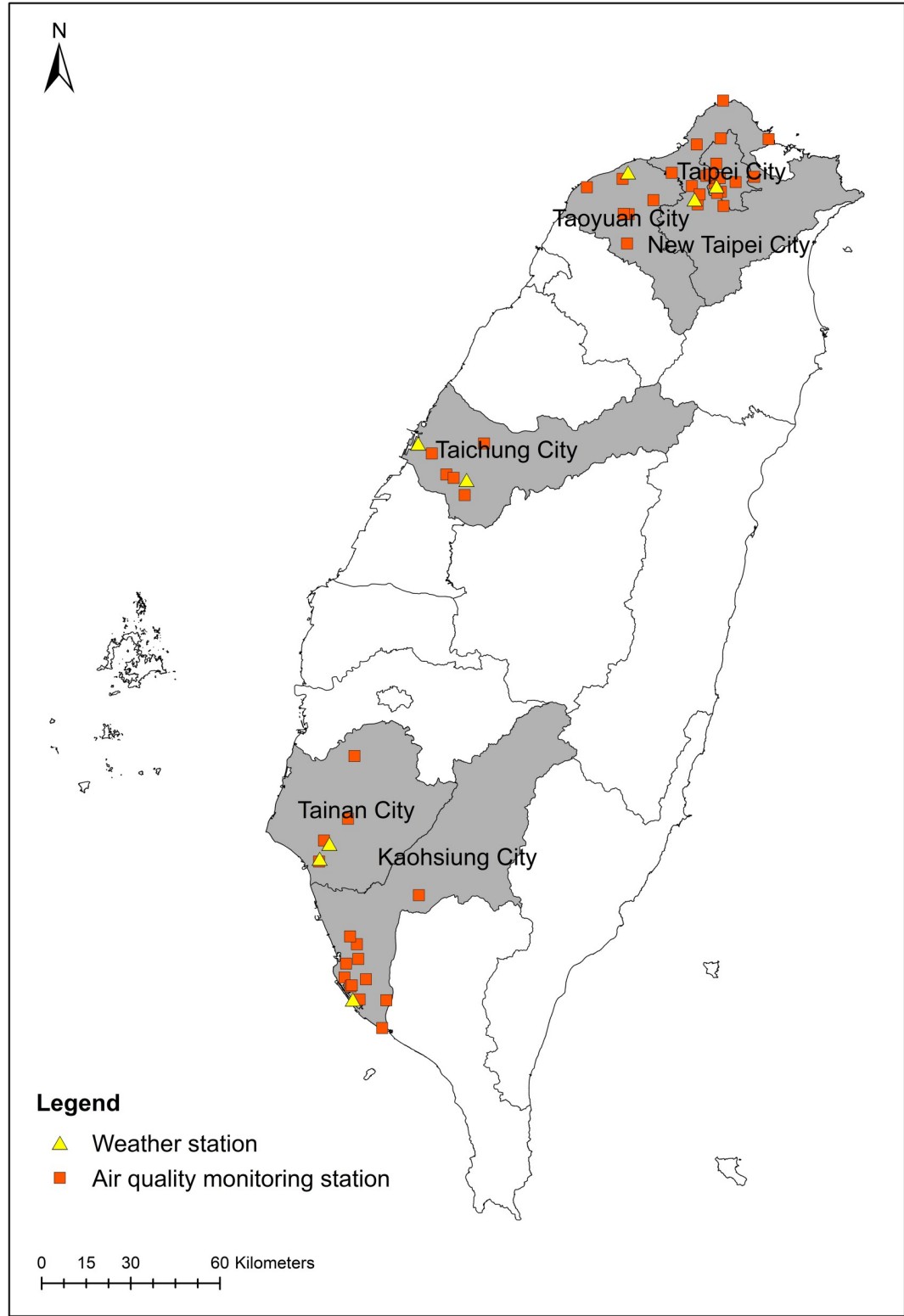

**Fig 1. Locations of weather observatories and ambient air quality monitor stations in metropolitan cities in Taiwan.**

outpatient visits. In temperature-health association analysis, this study divided the age groups into 0–18 years, 19–39 years, 40–64 years, and 65 years and above.

This was calculated as:

$$\mathbf{Log[Y] \sim BS(T, lag) + BS(PM_{2.5} \ or \ O_3, lag5) + NS(date, 7 \ per \ year) + NS(ws, 5)} \\ \mathbf{+ NS(rh, 4) + holiday + dow + PI}$$

Here, Y is the age-sex-cause specific daily number of deaths, ERVs, or outpatient visits; and T is area-specific daily mean temperature. We set temperature-health risk association as the basis spline (BS) function with four degrees of freedom (*df*) for the daily mean temperature. The lag effects of temperature were estimated for lag 0–25 days for mortality [31, 32], lag 0–3 days for ERVs [33, 34], and lag 0–7 days for outpatient visits [35]. Daily $PM_{2.5}$ and $O_3$ concentrations were also set as the BS function with four *df* and the lag day was set at 5-days [36]. Daily wind speed (ws) and relative humidity (rh) were included in the model as confounder and were set as the natural spline (NS). A variable day of week (*dow)* was included to control confounding weekly pattern. We also included daily case numbers of mortality from and morbidity of pneumonia and influenza (PI) as covariates.

The age-sex-cause specific pooled relative risk (RR) and 95% confidence intervals (CIs) of deaths, ERVs, or outpatient visits associated with daily mean temperatures for six major metropolitan cities in Taiwan were estimated using multivariate meta-analysis in comparison with the optimum temperature (i.e. the temperature with the least risk of mortality and morbidity) [37]. In addition, the RR and 95% CIs of each age-sex-cause specific mortality and morbidity by the increment of $PM_{2.5}$ relative to 17 μg/m$^3$ $PM_{2.5}$ (Q1) and by the increment of $O_3$ relative to 21 ppb (Q1), with and without adjusting temperature, were estimated at 90[th] percentile measurements. The model selection was based on lower Akaike information criterion [38].

Meta-analysis was fitted using a random-effects model by restricted maximum likelihood (REML). We computed all analyses using the *mgcv*, *dlnm*, and *mvmeta* package in R (version 3.3.3).

## Results

### Trends of mortality and morbidity and climatic characteristics from 2005 to 2016 in metropolitans in Taiwan

In the 6 metropolitans, during 2005–2016, the death numbers were greater from COPD than from asthma for both daily average deaths (7 cases versus 1 case) and the daily maximum deaths (23 versus 7 cases). The daily prevalence numbers of ERVs and outpatient visits were much greater in patients with asthma than in patients with COPD (Table 1). Young population was more likely to make outpatient visits up to 3,222 cases per day (S1 Fig in S1 File).

Table 1 also shows that the daily mean temperature was 23.7˚C (range: 5.2–33.0˚C), with a mean relative humidity of 74.9%, wind speed of 5.1 m/s, $PM_{2.5}$ concentrations of 30.6 μg/m$^3$, $O_3$ concentration of 28.4 ppb. Temperatures of <15˚C appeared in cold months from January to March, whereas temperatures of >27˚C occurred in the hottest months from June to August (S2 Fig in S1 File). The $PM_{2.5}$ concentrations were reversely associated with the temperatures, greater in cold days.

### Relative risk of cause-specific mortality, ERVs, and outpatient visits associated with ambient temperature, concentration of $PM_{2.5}$ and $O_3$

Fig 2 displays the pooled RRs of cause-specific mortality associated with daily mean temperatures over a lag of 0–25 days for subpopulation of the elderly. The temperature associated with

**Table 1. Means and ranges of daily cause-specific deaths, ERVs, outpatient visits and ambient environment conditions in six metropolitans in Taiwan from 2005 to 2016.**

| | Mean | Minimum | Q1 | Q2 | Q3 | Maximum |
|---|---|---|---|---|---|---|
| **Vital statics** | | | | | | |
| Asthma | 1 | 0 | 0 | 1 | 2 | 7 |
| COPD | 7 | 0 | 5 | 7 | 9 | 23 |
| **Emergency Room Visits** | | | | | | |
| Asthma | 172 | 75 | 129 | 151 | 187 | 1236 |
| COPD | 42 | 12 | 35 | 41 | 48 | 113 |
| **Outpatient visits** | | | | | | |
| Asthma | 14272 | 4461 | 11254 | 13785 | 16601.5 | 60991 |
| COPD | 800 | 38 | 394 | 936 | 1090 | 1612 |
| **Environmental factors** | | | | | | |
| Temperature (°C) | 23.7 | 5.2 | 19.6 | 24.8 | 28.3 | 33.0 |
| Relative humidity (%) | 75.0 | 29.1 | 70.0 | 75.0 | 80.5 | 100.0 |
| Wind speed (m/s) | 5.1 | 0.1 | 1.9 | 2.8 | 4.9 | 53.1 |
| $PM_{2.5}$ (μg/m$^3$) | 30.6 | 2.7 | 17.2 | 26.3 | 40.4 | 144.5 |
| $O_3$ (ppb) | 28.4 | 1.9 | 20.8 | 27.1 | 34.8 | 82.9 |

Daily cause-specific deaths, ERVs, and outpatient visits were calculated for all age groups including 0–18 years, 19–39 years, 40–64 years, and 65 years and above

the lowest risk of mortality from asthma was 26°C and that from COPD was 24°C. Further analysis showed that the risk of deaths from COPD was significant for the elderly men when the temperature dropped to 14.1°C, with a RR of 1.38 (95% CI 1.05–1.80) (S1 Table in S1 File).

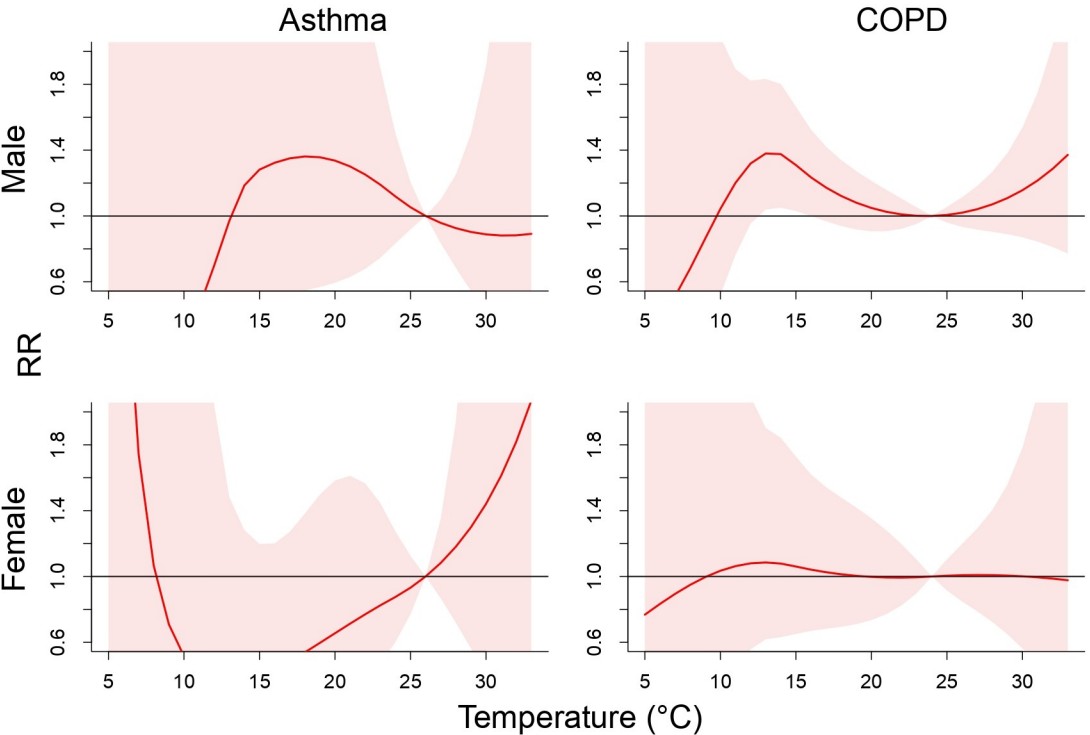

**Fig 2. Cumulative 26-day (lag 0–25) relative risk (95% confidence interval) of sex-specific deaths for asthma and COPD in the elderly associated with daily average temperature.**

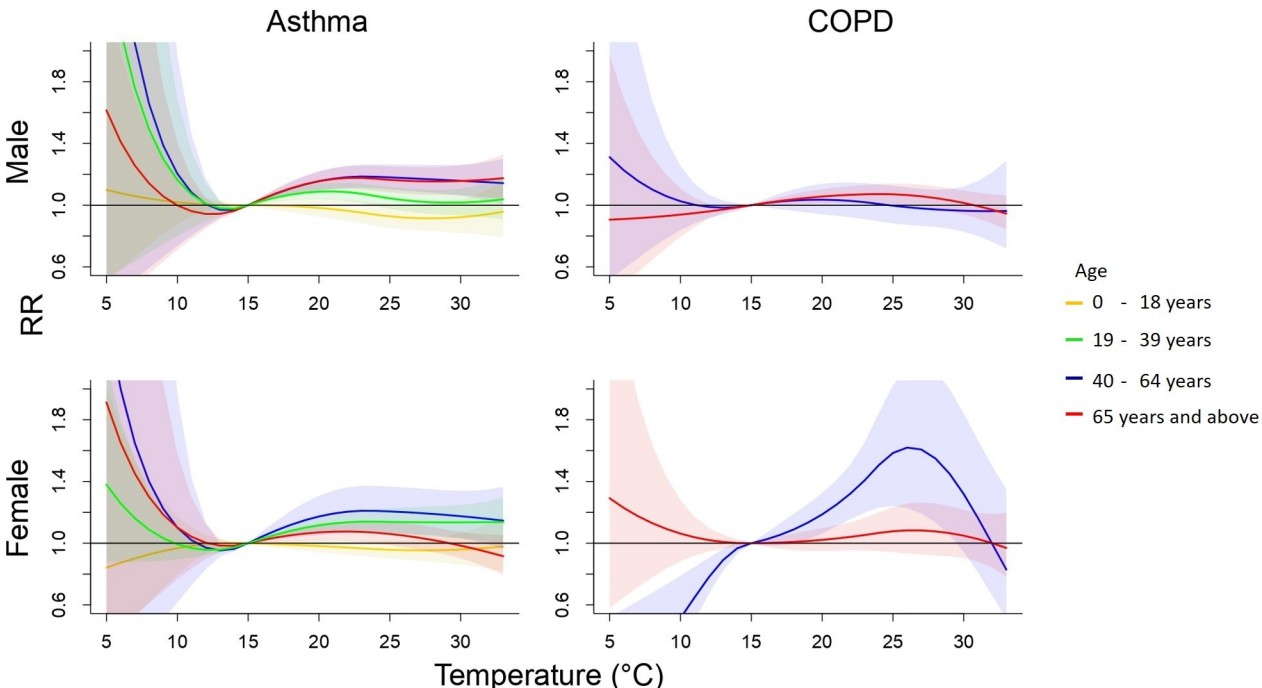

**Fig 3. Cumulative 4-day (lag 0–3) relative risk (95% confidence interval) of sex-age specific ERVs for asthma and COPD associated with daily average temperature.**

Sex-age specific analysis showed that men of 40–64 years old were at an elevated risk of mortality from COPD in high temperatures environment (S3 Fig in S1 File).

Fig 3 also shows the pooled cumulative 4-day RRs of cause-specific ERVs associated with daily mean temperatures by sex and age. The lowest risk of ERVs for asthma was estimated at 15°C. In addition, asthma has positive association with extreme high temperature in subpopulation men of 40–64 years old with a RR of 1.15 (95% CI 1.05–1.27) and 65 years and above with a RR of 1.16 (95% CI 1.06–1.27). The risk of extreme high temperature associated with asthma for subpopulation women of 19–39 years old is RR 1.14 (95% CI 1.02–1.26) and 40–64 years old with a RR 1.17 (95% CI 1.01–1.34) (S2 Table in S1 File). Women aged 40–64 years old were at an elevated risk of ERVs for COPD at a high temperature of 27°C.

Fig 4 shows the pooled cumulative 8-day RRs of cause-specific outpatient visits associated with daily mean temperature are shown by sex and age. The lowest risks of outpatient visits were identified at 28°C for asthma, and 16°C for COPD. Young population was at an elevated risk of outpatient visits for asthma at low temperatures with a RR 1.34 (95% CI 1.22–1.47) in male and RR 1.54 (95% CI 1.47–1.60) in female (S3 Table in S1 File).

Figs 5 and 6 display relative risks of outpatient visits associated with the daily mean $PM_{2.5}$ and $O_3$ concentrations relative to the Q1 levels, 17 $\mu g/m^3$ and 21 ppb, respectively, estimated by meta-analysis after adjusting for daily temperature. S6 Table in S1 File details RRs of visits for asthma by sex and age. Young population was at higher risks for asthma visits, the highest in male children after exposure to the 90th percentile of $PM_{2.5}$ level with an adjusted RR of 1.15 (95% CI 1.09–1.22). We also found positive association between daily $O_3$ concentration and asthma where younger population was at higher risk. The highest RR was observed in ERVs of asthma for female aged 40–64 years with a RR 1.21 (95% CI 1.05–1.39) (S8 Table in S1 File).

S4-S7 Figs in S1 File show the pooled RRs of cause-specific mortality associated with daily $PM_{2.5}$ concentrations for male subpopulations, ERVs associated with daily $PM_{2.5}$

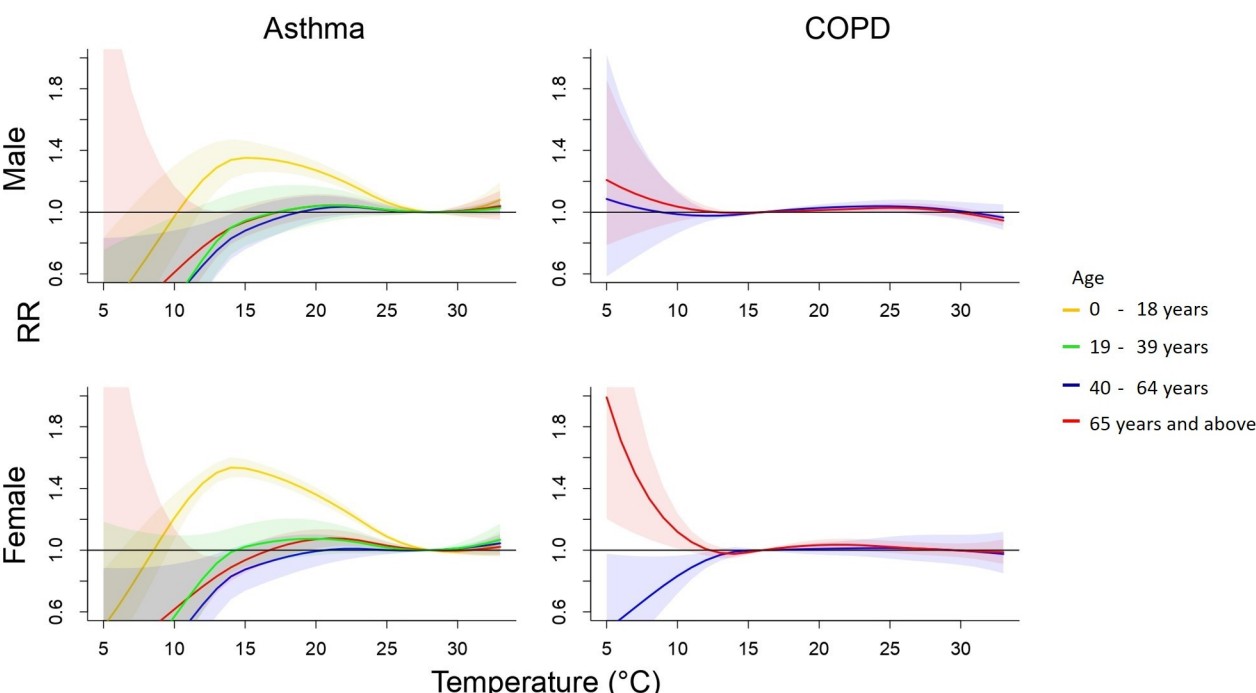

**Fig 4. Cumulative 8-day (lag 0–7) relative risk (95% confidence interval) of sex-age specific outpatient visits for asthma and COPD associated with daily average temperature.**

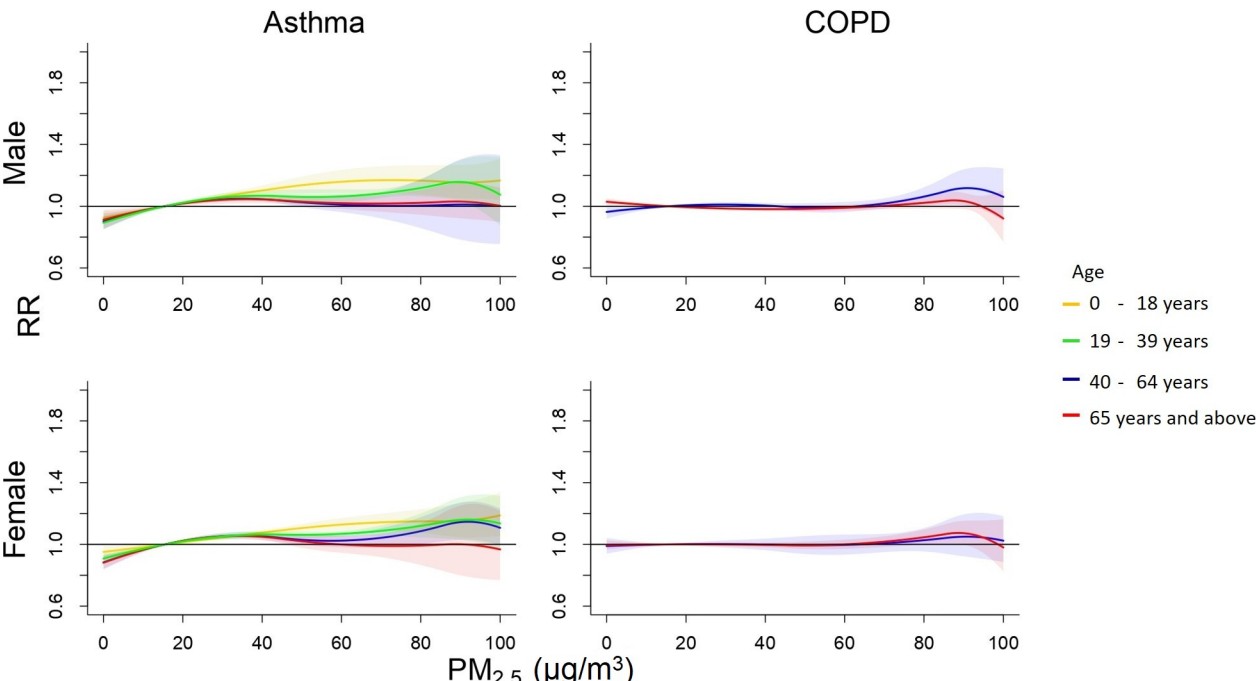

**Fig 5. Cumulative 6-day (lag 0–5) relative risk (95% confidence interval) of outpatient visits for asthma associated with daily PM$_{2.5}$ level relative to Q1 level (17 μg/m$^3$) after adjusting for daily average temperature.**

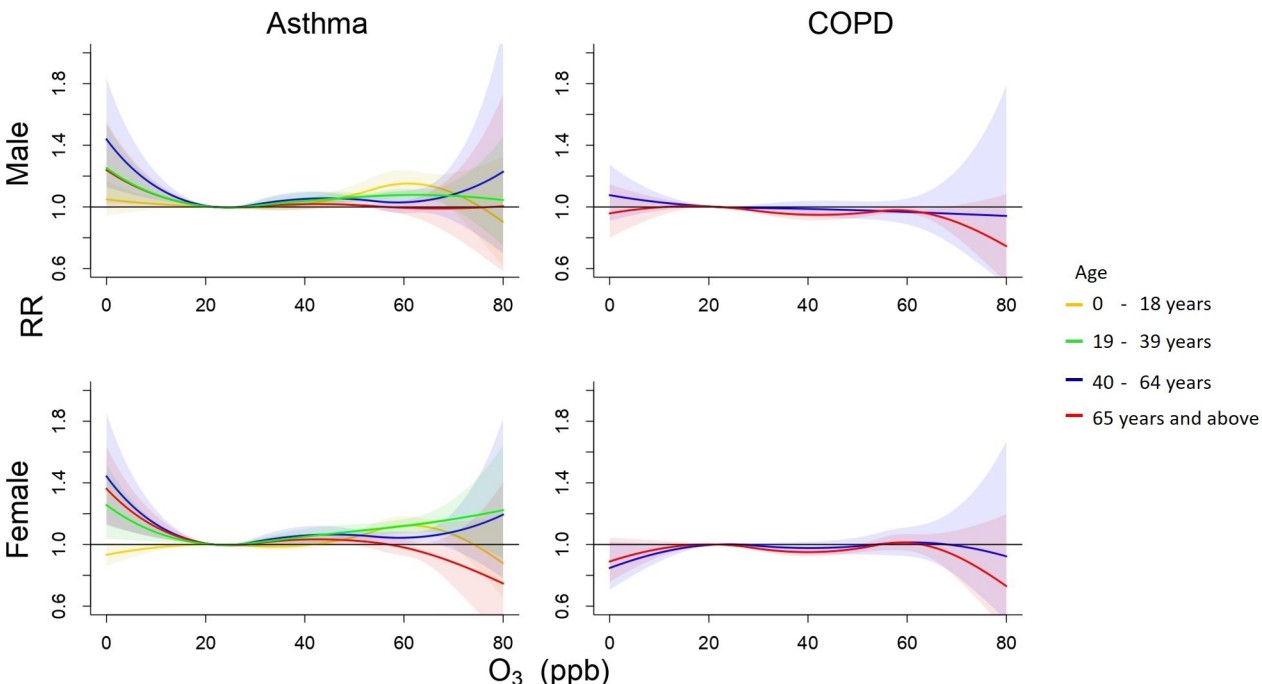

**Fig 6. Cumulative 6-day (lag 0–5) relative risk (95% confidence interval) of outpatient visits for COPD associated with daily O₃ level relative to Q1 level (21 ppb) after adjusting for daily average temperature.**

concentrations, mortality associated with daily $O_3$ concentrations, ERVs associated with daily $O_3$ concentrations. Risk estimates for cause-specific mortality, ERVs, and outpatient visits associated with daily 90[th] percentile $PM_{2.5}$ (55 μg/m³) and $O_3$ (43 ppb) relative to the Q1 levels (17 μg/m³ and 21 ppb, respectively) adjusting for daily temperatures are displayed in S4-S9 Tables in S1 File. S9 Table in S1 File show that the highest $O_3$ level without adjusting temperature associated with increased outpatient visits for asthma in children male population aged 0–18 years old with a RR 1.17 (95% CI 1.10–1.25), but not associated with outpatient visits for COPD.

## Discussion

This population-based study investigated risks of mortality from and morbidity (ERVs and outpatient visits) of asthma and COPD associated with extreme temperatures (daily mean temperatures at 99[th] and 5[th] percentiles) and 90[th] percentile levels of $PM_{2.5}$ and $O_3$. Impacts of extreme temperatures, and $PM_{2.5}$ and $O_3$ on health vary with studied type of disease, age, sex, and disease outcome. The present study found that asthmatic episodes were associated with extreme temperature, $PM_{2.5}$ and $O_3$, stronger for younger population of both sexes at extreme low temperature (14.1˚C). High concentrations of $PM_{2.5}$ (55 μg/m³) and $O_3$ (43 ppb) were also associated with increased risks of ERVs and outpatient visits for asthma and ERVs for COPD. In addition, mortality from COPD was also elevated in the elderly men at the extreme low temperature. These findings are consistent with findings in a previous study in Taiwan, which reported that events requiring ambulance services had a stronger association with low temperatures than with PM [36]. The PM concentration relationship with health events can interact with temperature because PM levels are higher when the ambient is cold.

Asthma is one of the most common chronic non-communicable diseases [39]. The disease has differential diagnosis between the acute and the chronic. Most acute asthma patient

known without laboratory test or chest radiographs because the symptom of the patient easily diagnoses such as breathlessness, inability to speak more than sort phrases, use of accessory muscle, or drowsiness [40]. Otherwise, chronic asthma patients usually suffer from this chronic condition (long-lasting or recurrent) and the physical symptoms are not always present in asthma sufferers, and it is possible to have asthma without presenting any physical maladies during an examination [41]. So, chronic asthma patients should undergo several test such as spirometry test or bronchoprovocation test before they are diagnosed [42].

Epidemiological studies have demonstrated that patients with asthma are susceptible to extreme temperatures with effect mostly for morbidity events [21, 43–45], and less for mortality. A study in Shanghai found that the risk of hospitalization for asthma was stronger with a longer duration in cold days than in hot days [45]. An Australia study found the risk of emergency admissions for asthma, among 0–14 years old children, was significant for younger boys in heat, whereas older children were most vulnerable to cold effects [43]. Low temperature could enhance the spread and triggering the cross-infection in crowded environments. The sudden air cooling escalates inflammation and causes the airway to narrow, leading to the incident asthma [43]. A study in the US found that heat could activate the sensory nerves of the vagal bronchopulmonary C-fibers, inducing reflex bronchoconstriction after activation [21]. Proportion of specific airway resistance appears in patients with asthma increased by near 3-fold greater at a high temperature of 49˚C than at the room temperature. In Taiwan, the ambient temperature is rarely over 35˚C. Low temperatures are more likely to trigger asthma symptoms. The relationship between extreme temperatures and the risk of asthma was only observed for ERVs and outpatient visits event (S2, S3 Tables in S1 File).

Studies have evaluated the effects of $PM_{2.5}$ and $O_3$ on children with asthma. Agree with previous studies, we found that ambient $PM_{2.5}$ level had significant association with greater risk of asthma [46–48]. Several epidemiological reviews have explained the biological mechanism for association between exposure to $PM_{2.5}$ and asthma [13, 46]. Liu et al. stated that $PM_{2.5}$ may comprise toxic materials including acids, metals, and nitrates, derived from combustion. These components will be accumulated into lungs and lead to allergy and inflammation [49]. Ozone has also found to be associated with elevated asthma risk. The exposure to $O_3$ can promote the infection on airways that increases the risk of asthma exacerbations [50]. Ozone is originated by photochemical reactions of oxygen and volatile organic compounds. These components were primarily derived from motor vehicle emission and were mainly detected in most residential areas with heavy road traffic [46].

This study also identified a stronger asthma risk in younger subpopulation (S3, S5, S6 and S8, S9 Tables in S1 File). Wang and Lin (2015) have reported that younger population had higher outpatient visits for respiratory diseases at extreme low temperature with a cumulative 8-day RRs of 1.36. Asthma is a major non-communicable disease declining the lung function and causing the risk of fixed airflow obstruction in children and adult [51]. Evidence indicates that age is a substantial factor affecting the vulnerability of individuals, where cases aged under 18 years are at higher risk [47]. The possible explanation is the incompleteness of children's lung growth and younger age groups inhale more air per unit of body weight [47]. Moreover, children may have higher outdoor activities than elderly subgroup.

Previous studies have widely reported the association between ambient temperatures and the risk of COPD [52–54]. Our study found that patients with COPD were at greater risks of mortality and outpatient visits, especially for the elderly subpopulation. In addition, the risk of ERVs for subpopulations aged 40–64 years was also significantly associated temperature at 26˚C. Similarly, Zhao, et al. found that younger groups (<65 years) were more susceptible to extreme heat while effect of low temperature was more acute for the elderly group [37]. Exposure to extreme temperatures may elevate the growth of pulmonary vascular resistance and

thrombosis leading to COPD symptoms. Additionally, the population living in cold areas may experience the increase in airway neutrophils, macrophages, and the levels of respiratory inflammation [37].

A population-based study conducted in London found that ambient $PM_{2.5}$ was significantly associated with lower lung function and increased COPD prevalence [55]. A cross-sectional study in China also revealed that prolonged chronic exposure to $PM_{2.5}$ resulted in decreased lung function, increased emphysematous lesions and airway inflammation [56]. A previous study explained the exposure to $PM_{2.5}$ triggered inflammatory cell infiltration and hyperemia in the lung and enhanced the inflammatory cells in the Broncho alveolar lavage fluid [57]. A study from Iran showed that $O_3$ had a significant impact on the hospital admissions, the number of COPD-related hospital admission increased by 2% (95% CI: 0.8–3.1) per 10 μg/m$^3$ increase in $O_3$ [58]. Ding et al. found that meteorological factors and pollutants ($PM_{2.5}$ and $O_3$) increased ERVs for COPD in the elderly in Taiwan [12]. However, our data showed that high $O_3$ level was associated with increased outpatient visits for asthma in younger population, but not associated with outpatient visits for COPD (S9 Table in S1 File).

High temperatures are distinctly conducive to intense convection. So, particulate matter is transported quickly and effectively, allowing its accelerated dispersion, and thus decreasing local mass concentrations. In contrast, low temperatures and the temperature inversion layer caused by radiative cooling weakens convection in these circumstances, atmospheric PM remains suspended under the inversion layer, leading to higher atmospheric PM concentrations [59]. Previous study found that low temperature enhanced the effects of $PM_{2.5}$ on COPD and the effect of $PM_{2.5}$ with low temperature significantly increased the risk on the COPD morbidity burden [60, 61]. A Study conducted in China suggests the indoor temperature should be kept at least on average temperature at 18.2°C, which may reduce the symptoms of COPD patients [62]. The present study also found that mortality from COPD was elevated at extreme low temperatures, but not at high temperature. However, the risk of emergency room visits for asthma was significant at high temperatures. This may be related to the fact that Taiwan's average temperature has risen at 1.1–1.6°C, faster than the global average of 0.8°C [7], thus, the chances of individuals being exposed to high temperature are increasing.

This study holds several strengths. First, we conducted this study employing the population-based health Insurance claims data, which are known as one of the largest databases in the world with the coverage of 99% population. Second, we were able to stratify data analyses by age and sex to investigate vulnerable subpopulations. Third, we considered certain confounders—including effect of holiday, the day of the week, long-term trends, and risk associated with infectious pneumonia and influenza—in our data analysis models. Evidences of this study may play an important role to improve public health policies for adapting to extreme temperatures associated with the health risks. Moreover, with better understanding of the vulnerability of populations by age, sex and disease in association with extreme temperatures, the policy makers could develop appropriate strategies for future climate-change scenarios.

There are limitations in this study. The Taiwan's government has set a regulation and ambitious action to combat the extreme temperatures and air pollution. Taiwan has constantly enhanced its Air Pollution Control Act to lower air pollution and improve the energy transformation [63]. Yet, this study still observed hot and cold impacts on mortality and morbidities [64]. We were unable to justify the impacts of working hour, smoking, drinking and exercise on health events, because the information was unavailable from the claims data [65]. Exercise may trigger asthma attack [66, 67]. Children are more likely expose to pollution during exercise. Moreover, extreme temperatures and $PM_{2.5}$ and $O_3$ related mortality and morbidities could also be affected by local adaptation factors [68, 69]. Information on local adaptation measures were also unavailable.

## Conclusion

This study investigated temperature- and $PM_{2.5}$- and $O_3$-related risks of mortality from and ERVs and outpatient visits of asthma and COPD for residents living in Taiwan's metropolitan cities. Evidence suggested that the vulnerability against extreme temperatures, $PM_{2.5}$ and $O_3$ varied by the studied disease, age, sex, and health outcome. The elderly men are at an elevated risk of death from COPD associated with extreme low temperature. In contrast, adults are at elevated risk of episodes requiring emergency room visits for asthma associated with extreme high temperature. We also found subpopulations aged 0–18 years with asthma disease had significant risk associations with ambient levels of $PM_{2.5}$ and $O_3$. This study suggests that vulnerable populations should be cautious about taking adaptive actions during extreme ambient events.

## Supporting information

**S1 File.**
(DOCX)

## Acknowledgments

We would like to thank the Ministry of Health and Welfare, Environmental Protection Administration (EPA) and Central Weather Bureau, Executive Yuan for providing research data. Interpretations and conclusions herein do not represent the views of these agencies. The views or opinions expressed in this article are those of the writers and should not be construed as opinions of the Taiwan EPA. Mention of trade names, vendor names, or commercial products does not constitute endorsement or recommendation by Taiwan EPA.

## Author Contributions

**Conceptualization:** Yasmin Zafirah, Yu-Kai Lin, Yu-Chun Wang.

**Formal analysis:** Yasmin Zafirah, Li-Wen Deng, Yu-Chun Wang.

**Writing – original draft:** Yasmin Zafirah, Yu-Kai Lin, Yu-Chun Wang.

**Writing – review & editing:** Gerry Andhikaputra, Fung-Chang Sung, Yu-Chun Wang.

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
