## [Decision Letter · Decision Letter 0]

19 Apr 2021

PONE-D-21-06176

Mortality and Morbidity of Asthma and Chronic Obstructive Pulmonary Disease Associated with Ambient Environment in Metropolitans in Taiwan

PLOS ONE

Dear Dr. Yu-Chun Wang,

Thank you for submitting your manuscript to PLOS ONE. After careful consideration, we feel that it has merit but does not fully meet PLOS ONE’s publication criteria as it currently stands. Therefore, we invite you to submit a revised version of the manuscript that addresses the points raised during the review process.

We look forward to receiving your revised manuscript.

Kind regards,

Won-Il Choi

Academic Editor

PLOS ONE

Journal Requirements:

2. We note that Figure 1 in your submission contain map images which may be copyrighted. All PLOS content is published under the Creative Commons Attribution License (CC BY 4.0), which means that the manuscript, images, and Supporting Information files will be freely available online, and any third party is permitted to access, download, copy, distribute, and use these materials in any way, even commercially, with proper attribution. For these reasons, we cannot publish previously copyrighted maps or satellite images created using proprietary data, such as Google software (Google Maps, Street View, and Earth). For more information, see our copyright guidelines: http://journals.plos.org/plosone/s/licenses-and-copyright.

2.1.    You may seek permission from the original copyright holder of Figure 1 to publish the content specifically under the CC BY 4.0 license. 

2.2.    If you are unable to obtain permission from the original copyright holder to publish these figures under the CC BY 4.0 license or if the copyright holder’s requirements are incompatible with the CC BY 4.0 license, please either i) remove the figure or ii) supply a replacement figure that complies with the CC BY 4.0 license. Please check copyright information on all replacement figures and update the figure caption with source information. If applicable, please specify in the figure caption text when a figure is similar but not identical to the original image and is therefore for illustrative purposes only.

Reviewers' comments:

Reviewer's Responses to Questions

**Comments to the Author**

1. Is the manuscript technically sound, and do the data support the conclusions?

Reviewer #1: Yes

Reviewer #2: Yes

2. Has the statistical analysis been performed appropriately and rigorously? 

Reviewer #1: Yes

Reviewer #2: Yes

3. Have the authors made all data underlying the findings in their manuscript fully available?

Reviewer #1: Yes

Reviewer #2: Yes

4. Is the manuscript presented in an intelligible fashion and written in standard English?

Reviewer #1: Yes

Reviewer #2: Yes

5. Review Comments to the Author

Reviewer #1: Authors investigated the association between extreme temperature, PM2.5 exposure and mortality and morbidity due to asthma and COPD by sex and age groups. They found that the mortality from COPD was significantly associated with extreme low temperature in elderly men, high PM2.5 exposure was associated with outpatient visit for asthma in boys, and high exposure of ozone was associated with ER visit due to asthma in female aged 40-64 years.

It is interesting that they found vulnerable populations for the extreme temperatures and ambient air pollutants in terms of mortality and morbidity due to chronic respiratory diseases.

Major

1. Authors included acute and chronic bronchitis diagnosis in the asthma definition. This may be the reason why numbers of asthma outpatients and ER visits were high and influence the results. What if asthma was defined excluding bronchitis?

2. They include ICD-10 J44 as COPD definition. Why ICD-10 J43 was not included?

3. For ozone concentration, did authors used 24-h average concentration or daytime concentration? This should be clarified.

4. Major finding of this paper is that low temperature was associated with increased mortality from COPD in elderly men. Authors mentioned this finding in line 217-222, 279-284, but did not suggest any mechanism. Is it because PM2.5 increased when the temperature is low?

5. In conclusion, authors mentioned that episodes of asthma in children are associated with the exposure to PM2.5 when it is cold and to O3 when it is hot. But they showed the results after adjusting for the daily temperature. I don’t find results stratified by cold or hot season. This should be explained.

Minor

1. In the abstract section, information about study population would be provided under methods sub-title.

2. In the introduction section, Line 62-67 is not related to this paper because they did not investigate the association with prevalence or incidence of chronic respiratory diseases but mortality and morbidity due to the diseases.

3. Line 171, Fig 3 seems to be figure S3.

4. Some tables are cited only in the discussion section. They would be better to be mentioned in the results section.

Reviewer #2: Dear Authors,

The study utilizes a non-linear statistical model with lag-effects to evaluate air pollution and weather-associated health risks in a large population of Taiwan. In general, the study is well planned, well structured and well written. However, the manuscript could be improved and clarified to ensure that readers understand the study better.

The authors should clarify the following sections to avoid confusion:

Introduction

The authors investigate the relationships between ambient environment and health-effects over different time periods. Introduction section lacked a short paragraph about the importance of the short-term acute and long-term cumulative effects of air pollution. It has been previously demonstrated that numbers of emergency room visits and cardiac arrests are increased during smog outbreaks, whereas long term exposure to air pollution affects mortality from and morbidity of numerous other diseases. (i.e. 10.3978/j.issn.2072-1439.2016.01.19). Also, the risk of respiratory infections might be increased by the acute (10.1016/j.chemosphere.2015.12.082) or prolonged (doi.org/10.3390/v13040556) exposure to air pollution.

Line 102:

On what rationale did the authors divide the studied population into this specific age groups? Was that based on the results of the statistical analysis, the known epidemiology of lung diseases, or was the division arbitrary?

Lines 126-130:

Could the authors please clarify, how exactly they decided on the number of lag days? Was that based on the lowest Akaike information criterion, previous studies, or, again, arbitrary? In a previous study referenced in no. 32 the authors have also included the 5-day lag effects of PM for out-of-hospital cardiac arrest events. However, the rationale on which that exact number of days was included in the model remains unclear. In the context of short and long-term effects of air pollution the number of days included in the model seems to be of a great importance.

Results

In my opinion the results section is difficult to follow and the most important results are difficult to be noticed. And these are that the effects of ambient environment on morbidity are greater than on mortality, and the effects on asthma are greater than on COPD. Also, many of the calculated effects were weak or non-significant. This section could be improved for clarity.

Lines 205-207

One of the most important findings is that the risk of emergency room or outpatient visits for asthma can be increased by up to 20% due to extreme levels of ozone and up to around 10% by the extreme levels of PM2.5. These results are only to be found in the supplementary tables and, in my opinion, these findings should be emphasized a little bit more so the readers could find them more easily.

Discussion

Lines 220-221

The authors should admit, that the differences between the calculated effects of extreme temperatures and extreme air pollution could be (at least partially) influenced by the number of days included in the mathematical model or the type of model that was used. It would be interesting to investigate long-term effects of air pollution on COPD and asthma exacerbation in this population in another study.

Lines 169 and 280-281.

The study found that mortality from COPD was elevated at low temperatures. How the authors can explain this finding? The temperature of 14°C does not feel extremely low. Should not it be a convenient ambient temperature for people with chronic lung diseases? In a continental climate temperatures are much lower and average around 10°C with Q1 around 0°C.

Minor issues

Throughout the manuscript there were several typos, punctuation errors and phrasing issues. Also, in Figure 5 please change “ug” to “µg”.

Line 114: “…of fine particulates matter with diameters of 2.5 micron”. Should be: "of less than 2.5"

Discussion section – it is generally not welcome to include or reference results in the discussion section.

Kind regards,

(-)

6. PLOS authors have the option to publish the peer review history of their article (what does this mean?). If published, this will include your full peer review and any attached files.

Reviewer #1: **Yes: **Woo Jin Kim

Reviewer #2: **Yes: **Kacper Toczylowski

---

## [Author Response · Author response to Decision Letter 0]

13 May 2021

Journal Requirements:

2. We note that Figure 1 in your submission contain map images which may be copyrighted. All PLOS content is published under the Creative Commons Attribution License (CC BY 4.0), which means that the manuscript, images, and Supporting Information files will be freely available online, and any third party is permitted to access, download, copy, distribute, and use these materials in any way, even commercially, with proper attribution. For these reasons, we cannot publish previously copyrighted maps or satellite images created using proprietary data, such as Google software (Google Maps, Street View, and Earth). For more information, see our copyright guidelines: http://journals.plos.org/plosone/s/licenses-and-copyright.

2.1. You may seek permission from the original copyright holder of Figure 1 to publish the content specifically under the CC BY 4.0 license. 

Response: Thank you for your suggestion. First, we want to inform that we get map data from this website https://whgis.nlsc.gov.tw/English/5-1Files.aspx. The data in the website is open access, they give us authorization to use the data and we don’t need to make a permission to use the data. According to your suggestion, we write a caption in the figure legend. Please refer to figure legend: “Figure 1. Locations of weather observatories and ambient air quality monitor stations in metropolitan Cities in Taiwan. Reprinted from [https://whgis.nlsc.gov.tw/Opendata/Files.aspx] under a CC BY license, with permission from [National Land Surveying and Mapping Center], original copyright [2021].”

2.2. If you are unable to obtain permission from the original copyright holder to publish these figures under the CC BY 4.0 license or if the copyright holder’s requirements are incompatible with the CC BY 4.0 license, please either i) remove the figure or ii) supply a replacement figure that complies with the CC BY 4.0 license. Please check copyright information on all replacement figures and update the figure caption with source information. If applicable, please specify in the figure caption text when a figure is similar but not identical to the original image and is therefore for illustrative purposes only.

Reviewers' comments:

Reviewer's Responses to Questions

Comments to the Author

1. Is the manuscript technically sound, and do the data support the conclusions?

Reviewer #1: Yes

Reviewer #2: Yes

2. Has the statistical analysis been performed appropriately and rigorously?

Reviewer #1: Yes

Reviewer #2: Yes

3. Have the authors made all data underlying the findings in their manuscript fully available?

Reviewer #1: Yes

Reviewer #2: Yes

4. Is the manuscript presented in an intelligible fashion and written in standard English?

Reviewer #1: Yes

Reviewer #2: Yes

 

5. Review Comments to the Author

Reviewer #1: Authors investigated the association between extreme temperature, PM2.5 exposure and mortality and morbidity due to asthma and COPD by sex and age groups. They found that the mortality from COPD was significantly associated with extreme low temperature in elderly men, high PM2.5 exposure was associated with outpatient visit for asthma in boys, and high exposure of ozone was associated with ER visit due to asthma in female aged 40-64 years.

It is interesting that they found vulnerable populations for the extreme temperatures and ambient air pollutants in terms of mortality and morbidity due to chronic respiratory diseases.

Major

1. Authors included acute and chronic bronchitis diagnosis in the asthma definition. This may be the reason why numbers of asthma outpatients and ER visits were high and influence the results. What if asthma was defined excluding bronchitis?

Response: Thank you for your comments. As we conducted the evaluation, we found daily case numbers of asthma (outpatients and ER visits) were not sufficient for the distributed lag nonlinear models analysis. Thus, after we reviewed prior studies in the similar field [1, 2], we decided to combine asthma, acute bronchitis and chronic bronchitis (ICD-9 466,491, 493 and ICD-10 J45-J46) in our assessment. 

1. Lam HC-y, Li AM, Chan EY-y, Goggins WB. The short-term association between asthma hospitalisations, ambient temperature, other meteorological factors and air pollutants in Hong Kong: a time-series study. 2016;71(12):1097-109. doi: 10.1136/thoraxjnl-2015-208054 %J Thorax.

2. Shoraka H, Soodejani M, Abobakri O, Khanjani N. The Relation between Ambient Temperature and Asthma Exacerbation in Children: A Systematic Review. Journal of Lung Health and Disease. 2019;3(1):1-9.

2. They include ICD-10 J44 as COPD definition. Why ICD-10 J43 was not included?

Response: Thank you for your comment. We add some reference in the method section to make it clearer. Please refer to lines 103-104: “Refer to previous Asian's reports [3, 4], this study only selected ICD-10 J44 as COPD definition and assess the associations with ambient environment”. 

3. Zhang Y, Wang Z, Cao Y, Zhang L, Wang G, Dong F, et al. The effect of consecutive ambient air pollution on the hospital admission from chronic obstructive pulmonary disease in the Chengdu region, China. Air Quality, Atmosphere & Health. 2021. doi: 10.1007/s11869-021-00998-9.

4. Jung HI, Park JS, Lee M-Y, Park B, Kim HJ, Park SH, et al. Prevalence of lung cancer in patients with interstitial lung disease is higher than in those with chronic obstructive pulmonary disease. Medicine. 2018;97(11).

3. For ozone concentration, did authors used 24-h average concentration or daytime concentration? This should be clarified.

Response: Yes. We address detailed information about using 24-h average concentrations for PM2.5 and ozone to make it clear. Please refer to lines 117-119: “We used 24-h average concentrations of fine particulates matter with diameters of less than 2.5 micron (PM2.5) and ozone (O3) in this study.”

4. Major finding of this paper is that low temperature was associated with increased mortality from COPD in elderly men. Authors mentioned this finding in line 217-222, 279-284, but did not suggest any mechanism. Is it because PM2.5 increased when the temperature is low?

Response: Thank you for your comment. It is true that the concentration of PM2.5 is higher in the winter of Taiwan. And possible synergistic adverse effects for low temperature and ambient PM2.5 concentrations on population health have also been addressed [5]. This study included both variables in distributed lag nonlinear models, and temperature related risks for mortality from COPD were estimated after controlling the effects of PM2.5. 

In addition, we add some discussion about the mechanism of low temperature effected the concentration of PM2.5. Please refer to lines 292-300: “High temperatures are distinctly conducive to intense convection. So, particulate matter is transported quickly and effectively, allowing its accelerated dispersion, and thus decreasing local mass concentrations. In contrast, low temperatures and the temperature inversion layer caused by radiative cooling weakens convection in these circumstances, atmospheric PM remains suspended under the inversion layer, leading to higher atmospheric PM concentrations [6]. Previous study found that low temperature enhanced the effects of PM2.5 on COPD, and the effect of PM2.5 with low temperature significantly increased the risk on the COPD morbidity burden [7, 8]. A Study conducted in China suggests the indoor temperature should be kept at least on average temperature at 18.2 °C that may reduce the symptoms of COPD patients [9]” 

5. McCormack MC, Paulin LM, Gummerson CE, Peng RD, Diette GB, Hansel NN. Colder temperature is associated with increased COPD morbidity. The European respiratory journal. 2017;49(6):1601501. doi: 10.1183/13993003.01501-2016. PubMed PMID: 28663313.

6. Li Y, Chen Q, Zhao H, Wang L, Tao R. Variations in PM10, PM2.5 and PM1.0 in an Urban Area of the Sichuan Basin and Their Relation to Meteorological Factors. 2015;6(1):150-63. PubMed PMID: doi:10.3390/atmos6010150.

7. Qiu H, Tan K, Long F, Wang L, Yu H, Deng R, et al. The Burden of COPD Morbidity Attributable to the Interaction between Ambient Air Pollution and Temperature in Chengdu, China. Int J Environ Res Public Health. 2018;15(3). Epub 2018/03/15. doi: 10.3390/ijerph15030492. PubMed PMID: 29534476; PubMed Central PMCID: PMCPMC5877037.

8. Duan R-R, Hao K, Yang T. Air pollution and chronic obstructive pulmonary disease. Chronic Diseases and Translational Medicine. 2020;6(4):260-9. doi: https://doi.org/10.1016/j.cdtm.2020.05.004.

9. Mu Z, Chen PL, Geng FH, Ren L, Gu WC, Ma JY, et al. Synergistic effects of temperature and humidity on the symptoms of COPD patients. Int J Biometeorol. 2017;61(11):1919-25. Epub 2017/06/02. doi: 10.1007/s00484-017-1379-0. PubMed PMID: 28567499.

5. In conclusion, authors mentioned that episodes of asthma in children are associated with the adjusting for the daily temperature. I don’t find results stratified by cold or hot season. This should be explained.

Response: Thank you for your comment. We apology if our conclusion did not match with the result. So, we have a little change on our conclusion. Please refer to lines 341-359:” This study investigated temperature- and PM2.5- and O3-related risks of mortality from and ERVs and outpatient visits of asthma and COPD for residents living in Taiwan’s metropolitan cities. Evidence suggested that the vulnerability against extreme temperatures, PM2.5 and O3 varied by the studied disease, age, sex, and health outcome. The elderly men are at an elevated risk of death from COPD associated with extreme low temperature. In contrast, adults are at elevated risk of episodes requiring emergency room visits for asthma associated with extreme high temperature. We also found subpopulations aged 0-18 years with asthma disease had significant risk associations with ambient levels of PM2.5 and O3. This study suggests that vulnerable populations should be cautious about taking adaptive actions during extreme ambient events.”

Minor

1. In the abstract section, information about study population would be provided under methods sub-title.

Response: Thank you for your comment. We already revised according to your suggestion. Please refer to lines 34-36: “The distributed lag non-linear model was employed to assess age (0-18, 19-39, 40-64, and 65 years and above), sex-cause-specific deaths, ERVs, and outpatient visits associated with extreme high (99th percentile) and low (5th percentile) temperatures and PM2.5 and O3 concentrations at 90th percentile.”

2. In the introduction section, Line 62-67 is not related to this paper because they did not investigate the association with prevalence or incidence of chronic respiratory diseases but mortality and morbidity due to the diseases.

Response: Thank you for your comment. We quoted the report from World Health Organization showing estimates of asthma and COPD in terms of mortality and morbidity to emphasize these growing vulnerable population should be concerned. Please refer to lines 63-65: “According to the report of WHO, asthma is estimated to cause about 250,000 annual deaths worldwide, and COPD will move from fifth leading cause of death in 2002 to fourth place in the ranking projected to 2030 worldwide [10].”

10. World Health Organization. Global Surveillance, Prevention and Control of Chronic Respiratory Diseases: A Comprehensive Report. World Health Organization, 2007.

6. World Health Organization. Global Surveillance, Prevention and Control of Chronic Respiratory Diseases: A Comprehensive Report. World Health Organization, 2007.

Thank you for your comment. 

3. Line 171, Fig 3 seems to be figure S3.

Response: Thank you for your comment. We’ve corrected the typo. Please refer to line 177.

4. Some tables are cited only in the discussion section. They would be better to be mentioned in the results section.

Response: Thank you for your comment. According to your suggestion, we address more detailed information about our findings that shown in supplementary tables in the result section. The interpretations for supplementary Tables 2-9 have been moved to results. 

Please refer to lines 183-187, “In addition, asthma has positive association with extreme high temperature in subpopulation men of 10-64 years old with a RR of 1.15 (95% CI 1.05-1.27) and 65 years and above with a RR of 1.16 (95% CI 1.06-1.27). The risk of extreme high temperature associated with asthma for subpopulation women of 19-39 years old is RR 1.14 (95% CI 1.02-1.26) and 40-64 years old with a RR 1.17 (95% CI 1.01-1.34) (Supplementary Table. S2).”; 

Lines 193-195: “Children were at an elevated risk of outpatient visits for asthma at low temperatures with a RR 1.34 (95% CI 1.22-1.47) in male and RR 1.54 (95% CI 1.47-1.60) in female (Supplementary Tables. S3).”

Lines 212-221: “Supplementary Figs. S4–S7 show the pooled RRs of cause-specific mortality associated with daily PM2.5 concentrations for male subpopulations, mortality associated with daily PM2.5 concentrations for female subpopulations, ERVs associated with daily PM2.5 concentrations, mortality associated with daily O3 concentrations, ERVs associated with daily O3 concentrations. Risk estimates for cause-specific mortality, ERVs, and outpatient visits associated with daily 90th percentile PM2.5 (55 μg/m3) and O3 (43 ppb) relative to the Q1 levels (17 μg/m3 and 21 ppb, respectively) adjusting for daily temperatures are displayed in Supplementary Tables S4-S9. Supplementary Table S9 show that the highest O3 level without adjusting temperature associated with increased outpatient visits for asthma in children male population aged 0-18 years old with a RR 1.17 (95% CI 1.10-1.25), but not associated with outpatient visits for COPD.” 

Reviewer #2: Dear Authors,

The study utilizes a non-linear statistical model with lag-effects to evaluate air pollution and weather-associated health risks in a large population of Taiwan. In general, the study is well planned, well-structured and well written. However, the manuscript could be improved and clarified to ensure that readers understand the study better.

The authors should clarify the following sections to avoid confusion:

Introduction

The authors investigate the relationships between ambient environment and health-effects over different time periods. Introduction section lacked a short paragraph about the importance of the short-term acute and long-term cumulative effects of air pollution. It has been previously demonstrated that numbers of emergency room visit and cardiac arrests are increased during smog outbreaks, whereas long term exposure to air pollution affects mortality from and morbidity of numerous other diseases. (i.e. 10.3978/j.issn.2072-1439.2016.01.19). Also, the risk of respiratory infections might be increased by the acute (10.1016/j.chemosphere.2015.12.082) or prolonged (doi.org/10.3390/v13040556) exposure to air pollution.

Line 102:

On what rationale did the authors divide the studied population into this specific age groups? Was that based on the results of the statistical analysis, the known epidemiology of lung diseases, or was the division arbitrary?

Response: Thank you for your comment. We divide the studied population into this specific age group according to the setting of previous studies [11-13]. Most previous studies divided the studied population into 0-18, 19-39, 40-64, and 65 years and above, and assess the associations between ambient environment and risks of asthma and COPD.

11. Alhanti BA, Chang HH, Winquist A, Mulholland JA, Darrow LA, Sarnat SE. Ambient air pollution and emergency department visits for asthma: a multi-city assessment of effect modification by age. Journal of Exposure Science & Environmental Epidemiology. 2016;26(2):180-8. doi: 10.1038/jes.2015.57.

12. Gass K, Klein M, Sarnat SE, Winquist A, Darrow LA, Flanders WD, et al. Associations between ambient air pollutant mixtures and pediatric asthma emergency department visits in three cities: a classification and regression tree approach. Environmental health : a global access science source. 2015;14:58-. doi: 10.1186/s12940-015-0044-5. PubMed PMID: 26123216.

13. Strosnider HM, Chang HH, Darrow LA, Liu Y, Vaidyanathan A, Strickland MJ. Age-Specific Associations of Ozone and Fine Particulate Matter with Respiratory Emergency Department Visits in the United States. Am J Respir Crit Care Med. 2019;199(7):882-90. Epub 2018/10/03. doi: 10.1164/rccm.201806-1147OC. PubMed PMID: 30277796.

Lines 126-130:

Could the authors please clarify, how exactly they decided on the number of lag days? Was that based on the lowest Akaike information criterion, previous studies, or, again, arbitrary? In a previous study referenced in no. 32 the authors have also included the 5-day lag effects of PM for out-of-hospital cardiac arrest events. However, the rationale on which that exact number of days was included in the model remains unclear. In the context of short and long-term effects of air pollution the number of days included in the model seems to be of a great importance.

Response: Thank you for your comment. In previous review studies [2, 14, 15], to consider mortality and morbidity displacement, the researchers estimated the air pollutants related short-term health risks using the various lag day setting, e.g. lag0-1, lag0-2, lag0-4, lag0-7 and lag0-13 were considered, and lag0-4 was an appropriate setting. 

In addition, the lag effects of short-term mortality and morbidity risks associated with ambient temperature had been evaluated before. Researchers used the lag0-21 to lag0-30 for mortality [16, 17], lag0-3 for emergency room visits [18], and lag0-7 for outpatient visits [19].

2. Shoraka H, Soodejani M, Abobakri O, Khanjani N. The Relation between Ambient Temperature and Asthma Exacerbation in Children: A Systematic Review. Journal of Lung Health and Disease. 2019;3(1):1-9.

/10/03. doi: 10.1164/rccm.201806-1147OC. PubMed PMID: 30277796.

14. Pascal M, Wagner V, Corso M, Laaidi K, Ung A, Beaudeau P. Heat and cold related-mortality in 18 French cities. Environment International. 2018;121:189-98. doi: https://doi.org/10.1016/j.envint.2018.08.049.

15. Shao M, Yu L, Xiao C, Deng J, Yang H, Xu W, et al. Short-term effects of ambient temperature and pollutants on the mortality of respiratory diseases: A time-series analysis in Hefei, China. Ecotoxicology and Environmental Safety. 2021;215:112160. doi: https://doi.org/10.1016/j.ecoenv.2021.112160.

16. Chen R, Yin P, Wang L, Liu C, Niu Y, Wang W, et al. Association between ambient temperature and mortality risk and burden: time series study in 272 main Chinese cities. BMJ. 2018;363:k4306. Epub 2018/11/02. doi: 10.1136/bmj.k4306. PubMed PMID: 30381293; PubMed Central PMCID: PMCPMC6207921 at www.icmje.org/coi_disclosure.pdf and declare: support from the National Natural Science Foundation of China, and China Medical Board Collaborating Programme for the submitted work; no financial relationships with any organisations that might have an interest in the submitted work in the previous three years; and no other relationships or activities that could appear to have influenced the submitted work.

17. Zhang Y, Liu X, Kong D, Fu J, Liu Y, Zhao Y, et al. Effects of Ambient Temperature on Acute Exacerbations of Chronic Obstructive Pulmonary Disease: Results from a Time-Series Analysis of 143318 Hospitalizations. International journal of chronic obstructive pulmonary disease. 2020;15:213-23. doi: 10.2147/COPD.S224198. PubMed PMID: 32099346.

18. Zhang Y, Yan C, Kan H, Cao J, Peng L, Xu J, et al. Effect of ambient temperature on emergency department visits in Shanghai, China: a time series study. Environmental Health. 2014;13(1):100. doi: 10.1186/1476-069X-13-100.

19. Zhao Y, Huang Z, Wang S, Hu J, Xiao J, Li X, et al. Morbidity burden of respiratory diseases attributable to ambient temperature: a case study in a subtropical city in China. Environ Health. 2019;18(1):89. Epub 2019/10/28. doi: 10.1186/s12940-019-0529-8. PubMed PMID: 31651344; PubMed Central PMCID: PMCPMC6814053.

Results

In my opinion the results section is difficult to follow and the most important results are difficult to be noticed. And these are that the effects of ambient environment on morbidity are greater than on mortality, and the effects on asthma are greater than on COPD. Also, many of the calculated effects were weak or non-significant. This section could be improved for clarity.

Lines 205-207

One of the most important findings is that the risk of emergency room or outpatient visits for asthma can be increased by up to 20% due to extreme levels of ozone and up to around 10% by the extreme levels of PM2.5. These results are only to be found in the supplementary tables and, in my opinion, these findings should be emphasized a little bit more so the readers could find them more easily.

Response: Thank you for your comment. According to your suggestion, we expanded our findings related to the effect of extreme high level of ozone and PM2.5 in the result section. The interpretations for supplementary Tables 2-9 have been moved to results. 

Please refer to lines 183-187, “In addition, asthma has positive association with extreme high temperature in subpopulation men of 10-64 years old with a RR of 1.15 (95% CI 1.05-1.27) and 65 years and above with a RR of 1.16 (95% CI 1.06-1.27). The risk of extreme high temperature associated with asthma for subpopulation women of 19-39 years old is RR 1.14 (95% CI 1.02-1.26) and 40-64 years old with a RR 1.17 (95% CI 1.01-1.34) (Supplementary Table. S2).”; 

Lines 193-195: “Children were at an elevated risk of outpatient visits for asthma at low temperatures with a RR 1.34 (95% CI 1.22-1.47) in male and RR 1.54 (95% CI 1.47-1.60) in female (Supplementary Tables. S3).”

Lines 212-221: “Supplementary Figs. S4–S7 show the pooled RRs of cause-specific mortality associated with daily PM2.5 concentrations for male subpopulations, mortality associated with daily PM2.5 concentrations for female subpopulations, ERVs associated with daily PM2.5 concentrations, mortality associated with daily O3 concentrations, ERVs associated with daily O3 concentrations. Risk estimates for cause-specific mortality, ERVs, and outpatient visits associated with daily 90th percentile PM2.5 (55 μg/m3) and O3 (43 ppb) relative to the Q1 levels (17 μg/m3 and 21 ppb, respectively) adjusting for daily temperatures are displayed in Supplementary Tables S4-S9. Supplementary Table S9 show that the highest O3 level without adjusting temperature associated with increased outpatient visits for asthma in children male population aged 0-18 years old with a RR 1.17 (95% CI 1.10-1.25), but not associated with outpatient visits for COPD.”

Discussion

Lines 220-221

The authors should admit, that the differences between the calculated effects of extreme temperatures and extreme air pollution could be (at least partially) influenced by the number of days included in the mathematical model or the type of model that was used. It would be interesting to investigate long-term effects of air pollution on COPD and asthma exacerbation in this population in another study.

Response: Thank you for the comment. This study listed the health risks associated with daily average temperature at 5th and 99th percentiles and levels of air pollutants at 90th percentile. Indeed, the number of days would be much less than that of daily average. The Taiwan population, located in subtropical and tropical climate zone, has been adapted to heat, thus, the daily average temperature at 5th and 99th are representative to the extremes temperature patterns for local population. Moreover, the 90th percentile measurements for air pollutants are also indicating a relative high exposure on the basis of local air quality. Thank you for the suggestion, we will assess long-term effects of air pollution on COPD and asthma exacerbation using this population based health data.

Lines 169 and 280-281.

The study found that mortality from COPD was elevated at low temperatures. How the authors can explain this finding? The temperature of 14°C does not feel extremely low. Should not it be a convenient ambient temperature for people with chronic lung diseases? In a continental climate temperatures are much lower and average around 10°C with Q1 around 0°C.

Response: Thank you for your comment. Taiwan is located in the sub-tropical and tropical climate. So the temperature 14°C can be categorized into cold or extremely low temperature. Furthermore, the installation rate of heater is low in Taiwan, and Taiwanese rarely turn on the heater in winters. Previous study in China suggested the indoor temperature should be kept at least on average temperature at 18.2 °C, which may reduce the symptoms of COPD patients. Please refer to lines 294-296.

Minor issues

Throughout the manuscript there were several typos, punctuation errors and phrasing issues. Also, in Figure 5 please change “ug” to “µg”.

Response: Thank you for your comment. We’ve corrected the typo and change the unit from “ug” to “µg”. Please refer to Figure 5. 

Line 114: “…of fine particulates matter with diameters of 2.5 micron”. Should be: "of less than 2.5"

Response: Thank you for your comment. We changed the terms from previous terms to “We used 24-h average concentration of fine particulates matter with diameters of less than 2.5 micron (PM2.5) and ozone (O3) in this study.” Please refer to line 117-119.

Discussion section – it is generally not welcome to include or reference results in the discussion section.

Response: Thank you for your comment. We included the result in the discussion section aims to make the reader easier to know our findings. 

Kind regards,

(-)________________________________________

6. PLOS authors have the option to publish the peer review history of their article (what does this mean?). If published, this will include your full peer review and any attached files.

Do you want your identity to be public for this peer review? For information about this choice, including consent withdrawal, please see our Privacy Policy.

Reviewer #1: Yes: Woo Jin Kim

Reviewer #2: Yes: Kacper Toczylowski

---

## [Decision Letter · Decision Letter 1]

1 Jun 2021

PONE-D-21-06176R1

Mortality and Morbidity of Asthma and Chronic Obstructive Pulmonary Disease Associated with Ambient Environment in Metropolitans in Taiwan

PLOS ONE

Dear Dr. Wang,

Thank you for submitting your manuscript to PLOS ONE. After careful consideration, we feel that it has merit but does not fully meet PLOS ONE’s publication criteria as it currently stands. Therefore, we invite you to submit a minor revised version of the manuscript that addresses the points raised during the review process.

We look forward to receiving your revised manuscript.

Kind regards,

Won-Il Choi

Academic Editor

PLOS ONE

Journal Requirements:

Reviewers' comments:

Reviewer's Responses to Questions

**Comments to the Author**

1. If the authors have adequately addressed your comments raised in a previous round of review and you feel that this manuscript is now acceptable for publication, you may indicate that here to bypass the “Comments to the Author” section, enter your conflict of interest statement in the “Confidential to Editor” section, and submit your "Accept" recommendation.

Reviewer #1: (No Response)

Reviewer #2: All comments have been addressed

2. Is the manuscript technically sound, and do the data support the conclusions?

Reviewer #1: Yes

Reviewer #2: Yes

3. Has the statistical analysis been performed appropriately and rigorously? 

Reviewer #1: Yes

Reviewer #2: Yes

4. Have the authors made all data underlying the findings in their manuscript fully available?

Reviewer #1: Yes

Reviewer #2: Yes

5. Is the manuscript presented in an intelligible fashion and written in standard English?

Reviewer #1: Yes

Reviewer #2: Yes

6. Review Comments to the Author

Reviewer #1: 1. Authors would address that asthma definition included acute and chronic diagnosis in Discussion section.

2. Generally, maximum daily mean of 8 hour averages of ozone concentration is known to affect health effects. This should be also addressed.

Reviewer #2: Dear Authors,

All my comments and suggestions were addressed and in my opinion the manuscript is ready to be published now.

7. PLOS authors have the option to publish the peer review history of their article (what does this mean?). If published, this will include your full peer review and any attached files.

Reviewer #1: No

Reviewer #2: **Yes: **Kacper Toczylowski

---

## [Author Response · Author response to Decision Letter 1]

3 Jun 2021

Review Comments to the Author

Reviewer #1: 1. Authors would address that asthma definition included acute and chronic diagnosis in Discussion section.

Response: Thank you for your comment. We add some references in the discussion section to make it clearer. Please refer to line 236-244: ”Asthma is one of the most common chronic non-communicable diseases [1]. This disease has differential diagnosis between the acute and the chronic. Most acute asthma patient known without laboratory test or chest radiographs because the symptom of the patient easily diagnoses such as breathlessness, inability to speak more than sort phrases, use of accessory muscle, or drowsiness [2]. Otherwise, chronic asthma patients usually suffer from this chronic condition (long-lasting or recurrent) and the physical symptoms are not always present in asthma sufferers, and it is possible to have asthma without presenting any physical maladies during an examination [3]. So, chronic asthma patients should undergo several test such as spirometry test or bronchoprovocation test before they are diagnosed [4].”

1. Shoraka H, Soodejani M, Abobakri O, Khanjani N. The Relation between Ambient Temperature and Asthma Exacerbation in Children: A Systematic Review. Journal of Lung Health and Disease. 2019;3(1):1-9.

2. Fergeson JE, Patel SS, Lockey RF. Acute asthma, prognosis, and treatment. J Allergy Clin Immunol. 2017;139(2):438-47. Epub 2016/08/25. doi: 10.1016/j.jaci.2016.06.054. PubMed PMID: 27554811.

3. Bousquet J, Bousquet PJ, Godard P, Daures JP. The public health implications of asthma. Bulletin of the World Health Organization. 2005;83(7):548-54. Epub 2005/09/24. PubMed PMID: 16175830; PubMed Central PMCID: PMCPMC2626301.

4. Krishna Sailaja A. An overall review on chronic asthma. International Journal Pharmaceutics and Drug Analysis. 2014;2(3):275-9.

2. Generally, maximum daily mean of 8 hour averages of ozone concentration is known to affect health effects. This should be also addressed.

Response: Thank you for your comment. We already addressed that the analysis using 24-h average concentration of fine particulates matter (PM2.5) and ozone (O3) in the methodology section. Please refer to line 117-119:” We used 24-h average concentration of fine particulates matter with diameters of less than 2.5 micron (PM2.5) and ozone (O3) in this study. Refer to our previous study [5], 24-h average concentration of ozone is the preferred metric instead of 1-h maximum concentration and 8-h maximum concentration of ozone for assessing the association with outpatient visits for total respiratory diseases.” 

5. Lin Y-K, Chang S-C, Lin C, Chen Y-C, Wang Y-C. Comparing ozone metrics on associations with outpatient visits for respiratory diseases in Taipei Metropolitan area. Environmental Pollution. 2013;177:177-84. doi: https://doi.org/10.1016/j.envpol.2012.12.010.

Reviewer #2: Dear Authors,

All my comments and suggestions were addressed and in my opinion the manuscript is ready to be published now.

---

## [Editor Report · Decision Letter 2]

14 Jun 2021

Mortality and Morbidity of Asthma and Chronic Obstructive Pulmonary Disease Associated with Ambient Environment in Metropolitans in Taiwan

PONE-D-21-06176R2

Dear Dr. Wang,

We’re pleased to inform you that your manuscript has been judged scientifically suitable for publication and will be formally accepted for publication once it meets all outstanding technical requirements.

Kind regards,

Won-Il Choi

Academic Editor

PLOS ONE
---

## [Editor Report · Acceptance letter]

24 Jun 2021

PONE-D-21-06176R2 

Mortality and Morbidity of Asthma and Chronic Obstructive Pulmonary Disease Associated with Ambient Environment in Metropolitans in Taiwan 

Dear Dr. Wang:

I'm pleased to inform you that your manuscript has been deemed suitable for publication in PLOS ONE. Congratulations! Your manuscript is now with our production department. 

Kind regards, 

on behalf of

Dr. Won-Il Choi 

Academic Editor

PLOS ONE